# *"A man never cries"*: A mixed-methods analysis of gender differences in depression and alcohol use in Moshi, Tanzania

**Alena Pauley**[1], **Mia Buono**[1], **Madeline Metcalf**[1], **Kirstin West**[1], **Sharla Rent**[1,2], **William Nkenguye**[3], **Yvonne Sawe**[4], **Mariana Mikindo**[4], **Joseph Kilasara**[4,5], **Judith Boshe**[4,6], **Brandon A. Knettel**[1,7], **Blandina T. Mmbaga**[1,4,6], **Catherine A. Staton**[1,8]*

**1** Duke Global Health Institute, Duke University, Durham, North Carolina, United States of America, **2** Duke Department of Pediatrics, Duke University Medical Center, Durham, North Carolina, United States of America, **3** Kilimanjaro Christian Medical University College, Moshi, Tanzania, **4** Kilimanjaro Clinical Research Institute, Moshi, Tanzania, **5** Department of Clinical Nursing, Kilimanjaro Christian Medical University College, Moshi, Tanzania, **6** Kilimanjaro Christian Medical Centre, Moshi, Tanzania, **7** Duke School of Nursing, Duke University, Durham, North Carolina, United States of America, **8** Duke Department of Emergency Medicine, Duke University Medical Center, Durham, North Carolina, United States of America

* catherine.lynch@duke.edu

## Abstract

Globally, gender differences are well-documented in alcohol use behaviors and MDD, yet these remain understudied in Moshi, Tanzania. Understanding gender-specific nuances of these conditions is crucial for developing effective and culturally appropriate mental health treatments. This study aims to investigate gender differences in MDD, alcohol use, and other aspects of mental well-being among patients at Kilimanjaro Christian Medical Centre (KCMC). Six hundred and seventy-six patients presenting for care at the KCMC Emergency Department (ED) and Reproductive Health Centre (RHC) were enrolled between October 2021 and May 2022. Patients were selected through systematic random sampling and completed quantitative surveys, including the Alcohol Use Disorder Identification Test (AUDIT) and the Patient Health Questionnaire 9 (PHQ-9). Nineteen patients were purposively chosen from the study population for in-depth interviews (IDIs) exploring alcohol use, gender, and depression. ANOVA, chi-squared tests, adjusted log-binomial regressions, and a linear regression model were used to analyze quantitative data in RStudio. A grounded theory approach was used to analyze all IDIs in NVivo. Average [SD] PHQ-9 scores were 7.22 [5.07] for ED women, 4.91 [4.11] for RHC women, and 3.75 [4.38] among ED men. ED women held the highest prevalence of MDD (25%) compared to RHC women (11%) and ED men (7.9%) ($p < 0.001$). Depressive symptoms were associated with higher AUDIT scores for ED men ($R2 = 0.11$, $p < 0.001$). Qualitative analysis showed that while present for women, social support networks were notably absent for men, playing a role in alcohol use. For men, alcohol was described as a coping mechanism for stress. Intersectionality of gender, alcohol use, and depression is influenced by sociocultural and behavioral norms in Moshi. As such, multi-layered, gender-differentiated programming should be considered for the treatment of substance use and mental health conditions in this region.

**Data availability statement:** Data are only available upon reasonable request, as participants did not consent to public data publishing, and data transfer requires a written agreement approved by Kilimanjaro Christian Medical Centre Ethics Committee and the National Institute for Medical Research (Tanzania). Data inquiries can be sent to Gwamaka W. Nselela at gwamakawilliam14@gmail.com.

**Funding:** This project was funded by the Duke Global Health Institute Graduate Student funds (AMP), and the Josiah Trent Foundation (21-06 to CAS). These two financial awards funded the salaries of JK, YS, and MMi as research assistants hired specifically for this study. No other authors received specific funding for this work. Infrastructure built by NIH grants (R01 AA027512 and D43TW012205 to CAS) was used to support the data collection and analysis processed for this grant to understand gender-related aspects of alcohol use at KCMC. The funders had no role in study design, data collection and analysis, decision to publish, or preparation of the manuscript.

**Competing interests:** The authors have declared that no competing interests exist.

## Introduction

Alcohol is a significant contributor to the global burden of disease, having been directly linked to the onset of diabetes, certain cancers, sexually transmitted infections, cardiovascular and gastrointestinal diseases, as well as injuries and violence [1–3]. Another key but often understudied contributor to the alcohol-related burden of disease is its impact on mental health. Worldwide, over 95 million people have been diagnosed with alcohol use disorder (AUD), a highly disabling condition characterized by harmful drinking and the inability to stop or control one's drinking habits [4–6]. Depression is the most common mental health disorder to co-occur with AUD [7–11], and both depression and AUD commonly co-occur with other mental health conditions such as other substance abuse disorders, anxiety, dementia, and psychosis-related disorders. One-third of individuals diagnosed with AUD meet the criteria for major depressive disorder (MDD), and studies have shown that those with an AUD diagnosis have a 3.7 times greater risk of acquiring MDD [8,12,13]. On its own, depression is the leading cause of disability globally, and when AUD and MDD co-occur, the severity of depression is greater than when it occurs alone [9,12,14–16].

The intersection of mental health and AUD is poorly studied in many low- and middle-income countries (LMICs), where limited resources and a shortage of trained personnel are barriers for effective treatment [17–19]. This healthcare gap is especially concerning as alcohol use and AUD are rising in LMICs [12,20–22]. In Tanzania, for instance, there are only 55 psychologists and psychiatrists serving a nation of more than 63 million people [23].

Tanzania, which is situated on the Eastern coast of the African continent, reports especially high rates of alcohol use and AUD [24]. Though the prevalence of AUD is 5.1% globally and 3.7% within the World Health Organization's (WHO) Africa region, 6.8% of Tanzanians tested positive for AUD, and 20% partook in heavy episodic drinking [24,25]. In Moshi, a suburban town in Northern Tanzania, rising levels of unhealthy alcohol are influenced by its widespread availability, low cost, cultural norms that encourage early consumption, and deep integration of alcohol into sociocultural practices [26–29].

Gender is another important variable that can influence the presentation and rate of onset of both AUD and MDD, as well as alcohol use behaviors. Worldwide, men consume more alcohol, while women experience higher rates of depression. In 2016, for example, the worldwide average annual alcohol intake was 19.4 liters for men and 7.0 liters for women [24]. Similarly, in Tanzania, men are five times more likely to have AUD than women [24]. Conversely, MDD affects 3.0% of women but 1.8% of men worldwide, with women also bearing a greater share of depression-related disability [30]. When adjusting for age, women had a disability-adjusted life year (DALY) rate of 564 compared to 354 for men, indicating that women globally lost 210 more years of healthy life from depression-related disability or premature death than men [31].

In Moshi, gender norms significantly influence alcohol use behaviors. Social stigma against women drinking often results in lower consumption but also encourages more secretive drinking among those who do consume alcohol [32,33]. This stigma and the tendency toward private drinking hinder women's access to appropriate alcohol-related care. Stress has been identified as a significant contributor to drinking for both genders in Moshi, with relationship stress for women and financial stress for men facilitating increased alcohol consumption [32].

The prevalence of depression has been well-studied among specific patient groups in Moshi, including older adults [34], pregnant women [35], and people living with HIV [36]. Data on depression rates among other populations in Moshi, though, is scarce, and the intersection of depression, AUD, and gender is even less well-studied in this context. Since gender may be a key driver of AUD and mental health disorders, a better understanding of the role

it plays in MDD and AUD could help inform and design more effective treatment options, guide local health policy, and improve patient outcomes. This paper seeks to contribute to these long-term goals and build upon previous work in this region that has examined depression rates among specific patient populations [35,37,38], estimated rates of alcohol use and AUD [26,39], and explored gender differences in alcohol use in Moshi [33,40]. To achieve these aims, we (1) quantitatively compare gender differences in rates of MDD, (2) qualitatively explore the relationship between alcohol use and mental well-being, and (3) assess how this relationship varies according to gender among Emergency Department (ED) and Reproductive Health Center (RHC) patients at the Kilimanjaro Christian Medical Centre (KCMC).

## Methods

### Study design overview

Between October 2021 and May 2022, data were collected to explore gender differences in alcohol consumption in Moshi, Tanzania, an urban town of over 200,000 residents. This study employed a cross-sectional, mixed-methods, sequential explanatory approach, with all data collected at KCMC, a large referral hospital in the region that provides care to over 1.9 million people [41]. Data consisted of quantitative surveys and qualitative in-depth interviews (IDIs) with patients at KCMC's ED and RHC. These two sites were chosen to better understand harmful alcohol use behaviors in the ED and to provide a primarily female perspective from the RHC.

During the study period, 676 patients met inclusion criteria and were enrolled through a systematic random sampling strategy to ensure an unbiased representation of each patient population. Of those enrolled, 21 participants had incomplete surveys and were excluded from the quantitative analysis. The remaining 655 survey reports were used to determine the prevalence of MDD by gender and the association between MDD and AUD. Out of these enrolled patients, a purposive sampling approach was used to select individuals for participation in IDIs until saturation was reached. The qualitative findings from the final 19 participating individuals allowed for a deeper exploration of how gender impacts the overall relationship between mental health and risky alcohol use behaviors.

### Eligibility criteria

Participants were eligible for study participation if they were 18 years of age or older, fluent in KiSwahili, and receiving care at KCMC's RHC or ED. Participants were excluded from the study if they were unable to provide informed consent, or were not clinically sober, medically stabilized, or well enough to complete the survey verbally on their own.

### Quantitative data

**Sample size estimation.** The sample size for the quantitative analysis was determined based on the expected difference in Alcohol Use Disorder Identification Test (AUDIT) scores between the RHC and ED populations. Using insights from existing studies and input from local research team members, we estimated the prevalence of AUD to be 10% in women at the RHC, 15% in women at the ED, and 30% in men at the ED. With 80% power and 90% confidence, we calculated that a sample size of 587 participants would be sufficient to assess the prevalence of risky alcohol consumption across the three study groups. Since the anticipated difference in AUD prevalence was smaller between the two female populations than between males and females, the research team initially prioritized enrolling more women to ensure adequate statistical power for detecting differences between the female groups.

When preliminary analysis revealed that the AUD prevalence was considerably higher than expected (40% in ED women and 45% in ED men), we recalculated the required sample size. Based on these revised prevalence estimates, 1,200 participants would be needed to detect a meaningful difference in proportions between these groups. However, given constraints in study time and funding, it was not feasible to enroll this larger sample. With IRB approval, the study target was adjusted to recruit as many male and female participants as possible within the available timeline.

**Procedures.** All subjects were enrolled through a systematic random sampling strategy. Using ED and RHC intake triage registries, every third patient presenting for care was approached for study participation, with the exception of ED women. Every ED women patient was approached as KCMC's ED sees significantly more male patients than females. Tanzanian, Kiswahili-speaking research assistants provided a study overview to same-gendered patients in quiet, private clinic spaces once a patient had been medically stabilized and was clinically sober. If the patient was interested in participating, a thorough discussion of the study, including potential risks and benefits, ensued. Individuals were free to ask questions or decline participation. If the patient was willing to participate in the study, written consent was obtained, and survey questions were read orally.

**Instruments.** Quantitative surveys consisted of five main components: (1) the nine-item Patient Health Questionnaire (PHQ-9), (2) AUDIT, (3) the Drinkers' Inventory of Consequences (DrInC), (4) basic demographic data, and (5) self-reported alcohol use data consisting of scaled multiple choice or binary yes-no questions. The first and second components were developed through an extensive literature review and research team expertise developed through years of alcohol-based research in the region. The remaining survey components – AUDIT, DrInC, and PHQ-9 – are reliable and validated measures both globally and in the study setting [42–51]. The composite data collection tool was pilot-tested in the local language of Kiswahili, and appropriate changes were made to ensure the validation and cultural relevance of the survey instrument.

The Patient Health Questionnaire 9 (PHQ-9) was used to quantitatively assess depressive symptoms and estimate the prevalence of MDD among study participants. This survey tool, used globally as a clinical marker for depression, has been translated and clinically validated in Tanzania in the local language of Kiswahili [42–45]. The PHQ-9 is composed of 9 questions related to depressive symptoms, such as having poor appetite, poor sleep regulation, or thoughts of self-harm, with responses tallied on a Likert-type scale ranging from 0, "not at all," to 3, "nearly every day" over the previous two weeks. Scores thus range from 0 to 27, with higher scores indicating more severe depressive symptoms. Previous studies using the Kiswahili version of the PHQ-9 have determined 9 to be the optimal cut-off point for identifying a major depressive episode, meaning a patient has experienced five or more depressive symptoms for at least two weeks [42,46]. As such, scores of 9 or greater were of primary interest in this analysis to assess the prevalence of likely depression.

The Alcohol Use Disorder Identification Test (AUDIT) measures overall alcohol consumption and assesses for unhealthy alcohol use behaviors. AUDIT scores range from 0 to 40, with higher scores representing increasingly unsafe use. This survey tool was designed to flag patients who display signs of Harmful or Hazardous Drinking (HHD) [47–49]. HHD is defined as drinking habits that place a person at risk for experiencing negative health outcomes from their drinking, measured quantitatively as AUDIT scores greater than or equal to 8 [47–49,52]. AUDIT has been translated and cross-culturally validated at KCMC with a similar study population [49].

**Statistical analysis.** We quantitatively assessed the prevalence of MDD and HHD among ED male, ED female, and RHC female patients. PHQ-9 and AUDIT scores are presented as

means with standard deviations, except when dichotomized according to the recommended cut-off points to assess MDD and HHD. For PHQ-9, scores of 8 or less were labeled as 'no MDD,' and scores of 9 or more were characterized as 'MDD.' Similarly, for AUDIT, scores of 7 or less were labeled as 'no HHD,' and scores of 8 or more were characterized as 'HHD.' When dichotomized, HHD and MDD are presented as descriptive proportions. Continuous demographic, PHQ-9, and AUDIT variables underwent ANOVA testing to compare means across our multiple patient groups, while categorical variables were assessed through the Chi-squared test.

To explore the relationship between PHQ-9 and AUDIT scores, we first performed a linear regression model using PHQ-9 scores as the dependent variable and AUDIT scores as the independent variable. Given the frequent co-occurrence of depression and AUD in other LMIC regions, we sought to test this directionality while acknowledging the potential for a bilateral relationship. We also conducted a reverse regression model, treating AUDIT scores as the dependent variable and PHQ-9 scores as the independent variable. This allowed us to examine the robustness and bidirectionality of the relationship between depression and hazardous drinking within our study population.

Potential confounding factors, including age, employment, marital status, female gender, and clinical group, were also assessed in both models. Significant unadjusted predictors identified through preliminary analyses were further included in adjusted models to control for these confounders.

For the linear regression models, all assumptions were tested and met. Specifically, (1) there were no obvious departures from linearity as observed via scatter plot examination of the independent versus dependent variable, (2) independence of observations was assumed due to the nature of the sampling strategy, (3) homoscedasticity was assessed by plotting the residuals against the fitted values, confirming no discernable pattern, and (4) normality was evaluated and confirmed using inspection of the Normal Q-Q Plot, with residuals following the diagonal line. These assumptions were met only for the linear regression model.

Contingency tables, in combination with the "epiR" R package (version 2.0.19) [53], were used to calculate unadjusted relative risk (RR) for binary outcomes (e.g., MDD status) across independent binary predictors. Binary predictors of MDD status that were tested included female gender, clinical group, HHD status, and whether treatment was sought for alcohol use and/or psychiatric reasons. For adjusted RRs, we employed a generalized linear model with a log link (log-binomial regression) to control for confounding variables. Significant unadjusted predictors such as age, employment status, marital status, female gender, and clinical group, identified through preliminary analyses, were included as covariates in the adjusted model. The significance level for all statistical tests was set at $p < 0.05$, and two-tailed tests were applied.

Missing data were minimal for all variables analyzed (reported in Tables 1 and 2). Quantitative data were measured across all three patient groups instead of by gender alone to illustrate differences in MDD burden between the two clinical sites. Further, since the ED and RHC have distinct patients presenting for care, a delineation in scores more accurately described both populations. All quantitative data were analyzed in RStudio (version 1.4.1106) with standard and user-tested packages.

## Qualitative data

**Sampling strategy and sample size estimation.** Of the 655 individuals with fully completed surveys, the study team aimed to recruit up to 20 participants for participation in IDIs or until thematic saturation was reached, whichever came first. This dynamic approach

**Table 1. Patient Demographics.**

| Demographics by Population Type | Overall, N = 655 | ED Women, N = 271 | RHC Women, N = 270 | ED Men, N = 114 | P-Value |
|---|---|---|---|---|---|
| **Age**[1] | 42.2 (18.8) | 46.2 (22.8) | 36.4 (12.8) | 46.3 (17.0) | <0.001 |
| **Religion**[2] | | | | | 0.205 |
| None | 11 (1.7%) | 5 (1.8%) | 3 (1.1%) | 3 (2.5%) | |
| Muslim | 124 (18.6%) | 47 (17.3%) | 45 (16.5%) | 32 (26.2%) | |
| Christian | 529 (79.5%) | 218 (80.4%) | 224 (82.4%) | 87 (71.3%) | |
| Other | 1 (0.2%) | 1 (0.4%) | 0 (0.0%) | 0 (0.0%) | |
| Missing/Refused | | | | | |
| **Marital Status**[2] | | | | | <0.01 |
| Living w/ partner in registered marriage | 336 (50.5%) | 130 (48.0%) | 140 (51.5%) | 66 (54.1%) | |
| Living w/ partner in unregistered marriage | 78 (11.7%) | 25 (9.2%) | 41 (15.1%) | 12 (9.8%) | |
| Divorced/Separated | 32 (4.8%) | 16 (5.9%) | 7 (2.6%) | 9 (7.4%) | |
| Widowed | 80 (12.0%) | 45 (16.6%) | 27 (9.9%) | 8 (6.6%) | |
| Never Married | 137 (20.6%) | 55 (20.3%) | 57 (21.0%) | 25 (20.5%) | |
| Missing/Refused | | | | | |
| **Employment Status**[2] | | | | | <0.001 |
| Employed | 371 (56.6%) | 127 (46.9%) | 192 (71.1%) | 52 (45.6%) | |
| Unemployed | 215 (32.8%) | 111 (41.0%) | 50 (18.5%) | 54 (47.4%) | |
| Student | 69 (10.5%) | 33 (12.2%) | 28 (10.4%) | 8 (7.0%)8 (7.0%) | |

[1]ANOVA was performed for continuous variables; Mean (SD).

[2]Chi-square test was performed for categorical variables; n (%)

**Table 2. Study Population PHQ-9 and AUDIT Scores.**

| PHQ-9 and AUDIT Status by Population Type | Overall, N = 655[1] | ED Women, N = 271[1] | RHC Women, N = 270[1] | ED Men, N = 114[1] | P-Value |
|---|---|---|---|---|---|
| **PHQ-9 Score**[1] | 5.66 (4.77) | 7.22 (5.07) | 4.91 (4.11) | 3.75 (4.38) | <0.001 |
| **MDD Status (PHQ-9≥9)**[2] | 106 (16.2%) | 84 (31.0%) | 45 (16.7%) | 18 (15.8%) | <0.001 |
| **AUDIT Score**[1] | 3.22 (5.36) | 3.07 (4.76) | 1.86 (3.46) | 6.76 (8.16) | <0.001 |
| **HHD Status (AUDIT≥ 8)**[2] | 109 (16.6%) | 46 (17.0%) | 20 (7.4%) | 43 (37.7%) | <0.001 |
| **Sought Psychiatric Treatment**[2] | | | | | 0.895 |
| No | 592 (90.4%) | 245 (90.4%) | 242 (89.63%) | 105 (92.1%) | |
| Yes | 59 (9.0%) | 24 (8.9%) | 27 (10.0%) | 8 (7.1%) | |
| Missing/Refused | 4 (0.1%) | 2 (0.1%) | 1 (0.0%) | 1 (0.1%) | |

[1]T-test was performed for continuous variables; Mean (SD).

[2]Chi-square test was performed for categorical variables; n (%)

allowed us to ensure balanced representation while also leaving thematic saturation as the ultimate determinant for data collection. Our initial target sample size was based on widely accepted qualitative research guidelines recommending 15–20 interviews for saturation with narrow study aims and a relatively homogenous group [54]. Thematic saturation was defined as the lack of new themes or information after three consecutive interviews. To obtain a gender-balanced perspective, the team aimed to interview five female ED patients, five female RHC patients, and ten male ED patients, ceasing once saturation was reached, which occurred for ED males after nine interviews, leaving 19 IDIs collected in total.

Purposive sampling was employed to ensure a diverse demographic background and alcohol use experiences among selected participants. This included selecting participants from differing ages, religions, income groups, professions, and marital statuses and encompassing patients who held a diverse range of alcohol use behaviors, from complete abstinence to heavy use. Research assistants also worked to incorporate those with notable personal experiences with alcohol, such as those who used to drink heavily but quit or have had a close friend or relative drink excessively. To maintain a balanced and unbiased selection, interviewee characteristics were reviewed monthly by the study lead, and recruitment was revised accordingly.

**Procedures.** Selected participants were approached to participate in an IDI after the conclusion of the quantitative survey or through a subsequent phone call. Due to the potentially stigmatizing nature of the study topic, emphasis was placed on maintaining confidentiality for all participants. All IDIs were conducted face-to-face with the participant and the same gender-matched research assistant who had enrolled them previously. All interviews were audio-recorded and conducted in a private, unoccupied room at KCMC.

Interviews were conducted in Kiswahili using a semi-structured interview guide. The interview guide included open-ended questions with probes built in when the discussion was unclear, of particular interest, or inconsistent with previous statements. After an early review of quantitative and qualitative data, interesting trends around alcohol use, depression, and mental well-being arose. Thus, the research team amended the interview guide to include the following questions: "Is there a relationship between depression and alcohol use? If yes, please explain"; "Is the burden of alcohol use and depression higher among either men or women? Why or why not?" This change was implemented with IRB approval after the 7th IDI was conducted. Aside from these additions, the same interview guide was used for all participants. Participants were provided with refreshments halfway through the interview session and received a small stipend of 5,000 TSH (approximately 2.25 USD) for travel reimbursement.

**Analysis.** Qualitative interviews were analyzed using a grounded theory approach to allow new themes to arise organically and be fully explored [55]. Based on the first four interviews, a codebook was developed by the primary data analyst and the Tanzanian research team using both deductive and inductive coding schemes. To ensure cultural pertinence and maintain content validity, the codebook was consistently discussed and modified to include additional themes from new interviews.

The primary data analyst trained the Tanzanian research team on qualitative analysis and coding using NVivo 12 software. All team members coded the first three interviews individually and then discussed them to maintain coding reliability. If disagreements were present in coding schemes, the team discussed them until a consensus on the code was reached. This process continued until an 80% agreement was reached among the research team. After the analysis of the remaining 16 interviews, the analysis team approved the final coding, and content memos were created for each theme.

## Research ethics

Ethical approval for this study was obtained from the Duke University Institutional Review Board (Pro00107861), the Tanzanian National Institute of Medical Research (NIMR/HQ/R.8a/Vol. IX/3734), and the Kilimanjaro Christian Medical University College Ethical Review Board (No. 2515) prior to the onset of any data collection. Data were shared only through a data share agreement and were maintained in a de-identified manner as much as possible. Personal health information was used for screening and enrollment but was de-identified when data were stored and analyzed.

## Results

### Quantitative

Out of the 676 patients enrolled, 655 were analyzed. 21 participants were excluded from the quantitative analysis due to incomplete surveys. This study was composed primarily of female participants, most of whom were Christian (n = 529, 79.5%), were employed (n = 371, 56.6%), and were living with a partner either in a registered or unregistered marriage (n = 414, 62.2%) (Table 1). ED men were found to drink the highest quantities of alcohol, do so the most frequently, and spend the most money on alcohol each week of all three patient groups. The female RHC patient population had the youngest average age, which may be due to the fact that the RHC sees a large number of women who are of childbearing potential.

The highest average [SD] PHQ-9 score belonged to ED women (7.22 [5.07]), followed by RHC women (4.91 [4.11]) and ED Men (3.75 [4.38]) (Table 2). Across all patient populations, ED women also had the highest prevalence of MDD (31.0%), followed by RHC women (16.6%) and ED men (15.8%) (Table 2). Differences in average PHQ-9 scores and prevalence of MDD were statistically significant (p < 0.001) across the three patient groups (Table 2). More patients from the RHC women (10.0%) population had sought psychiatric treatment than ED women (8.9%) and ED men (7.1%), but this difference was not statistically significant (p = 0.895).

Female patients exhibited a statistically significant (p < 0.001) increased risk of MDD status, with a relative risk of 1.15 (95% CI: 1.07–1.23) (Table 3). This corresponds to a 15% increased risk of having MDD as compared to their male counterparts. In particular, women attending the ED (e.g., ED women) demonstrated a 71% increased risk of MDD compared to other clinical groups, with a relative risk of 1.71 (95% CI 1.43–2.06). In contrast, RHC Women and ED Men exhibited a significantly reduced risk of MDD by 33% (RR: 0.67, 95% CI: 0.49–0.92) and 58% (RR: 0.42, 95% CI: 0.22–0.79), respectively. Significant increases or decreases in the relative risk of MDD were not observed among individuals with HHD status or among those who sought psychiatric treatment (p > 0.1).

In the adjusted log-binomial regression model, after controlling for age, employment status, marital status, female gender, and clinical group, hazardous drinking (AUDIT ≥ 8) was not significant associated with MDD (RR = 1.02, 95% CI: 0.70–1.48, p = 0.92), whereas women continued to have a significant increased risk of MDD (RR = 2.31, 95% CI: 1.42–3.77, p < 0.01).

Finally, in examining the relationship between alcohol consumption and depressive symptoms, a positive linear association was observed between AUDIT and PHQ-9 scores

**Table 3. Univariate Relative Risks for Predictors of MDD Status.**

| Binary Predictors of MDD Status | MDD Prevalence (n/ N), (%) | Relative Risk (RR) (95% CI) | p-value |
|---|---|---|---|
| Female Gender | 129/ 541 (23.8%) | 1.10 (1.02–1.18) | <0.05 |
| Clinical Group | 147/ 655 (22.4%) | | |
| ED Women | 84/ 271 (31.0%) | 1.58 (1.32–1.90) | <0.01 |
| RHC Women | 45/ 270 (16.7%) | 0.70 (0.54–0.91) | <0.01 |
| ED Men | 18/ 114 (15.8%) | 0.61 (0.38–0.97) | <0.05 |
| HHD Status (AUDIT ≥ 8) | 109/ 655 (16.6%) | 1.07 (0.72–1.58) | 0.74 |
| Sought Psychiatric Treatment | 59/ 655 (9.0%) | 0.968 (0.91–1.03) | 0.43 |

Unadjusted univariate associations between predictors of MDD and MDD, reported as RR (95% CI).

*only* among ED men ($R^2 = 0.11$; $p < 0.001$) (Fig 1). This indicates that approximately 11% of the variance in PHQ-9 scores is explained by AUDIT scores in this group. No significant linear association was found between depression and alcohol use among ED women ($R^2 = 0$; $p < 0.668$) nor RHC women ($R^2 = 0.02$; $p < 0.014$). Data was missing only from the variable 'sought psychiatric treatment' analyzed in Tables 2 and 3, and these were excluded from the analysis.

## Qualitative

Nineteen patients (ED women, n = 5; RHC women, n = 5; ED men, n = 9) between the ages of 20 and 70 years participated in an IDI. While the majority were Christian (n = 15, 80%), education levels, relationship status, and alcohol use among IDI participants varied widely. Some participants held only primary-level education, while others held an advanced degree, were single, married, or divorced, and some never consumed alcohol while others had 3 to 4 drinks per sitting almost every day of the week.

Two overarching themes were deductively identified from the 19 completed IDIs: (1) the relationship between alcohol use and mental well-being and (2) gender imbalances in the effect of alcohol use on mental well-being (Table 4). Within the first theme, participants noted that alcohol consumption drove financial stress among men while also being used as a coping mechanism for stress. In the second theme, IDI participants noted a distinction in the strength of men versus women's social support networks used to deal with stressful situations. The

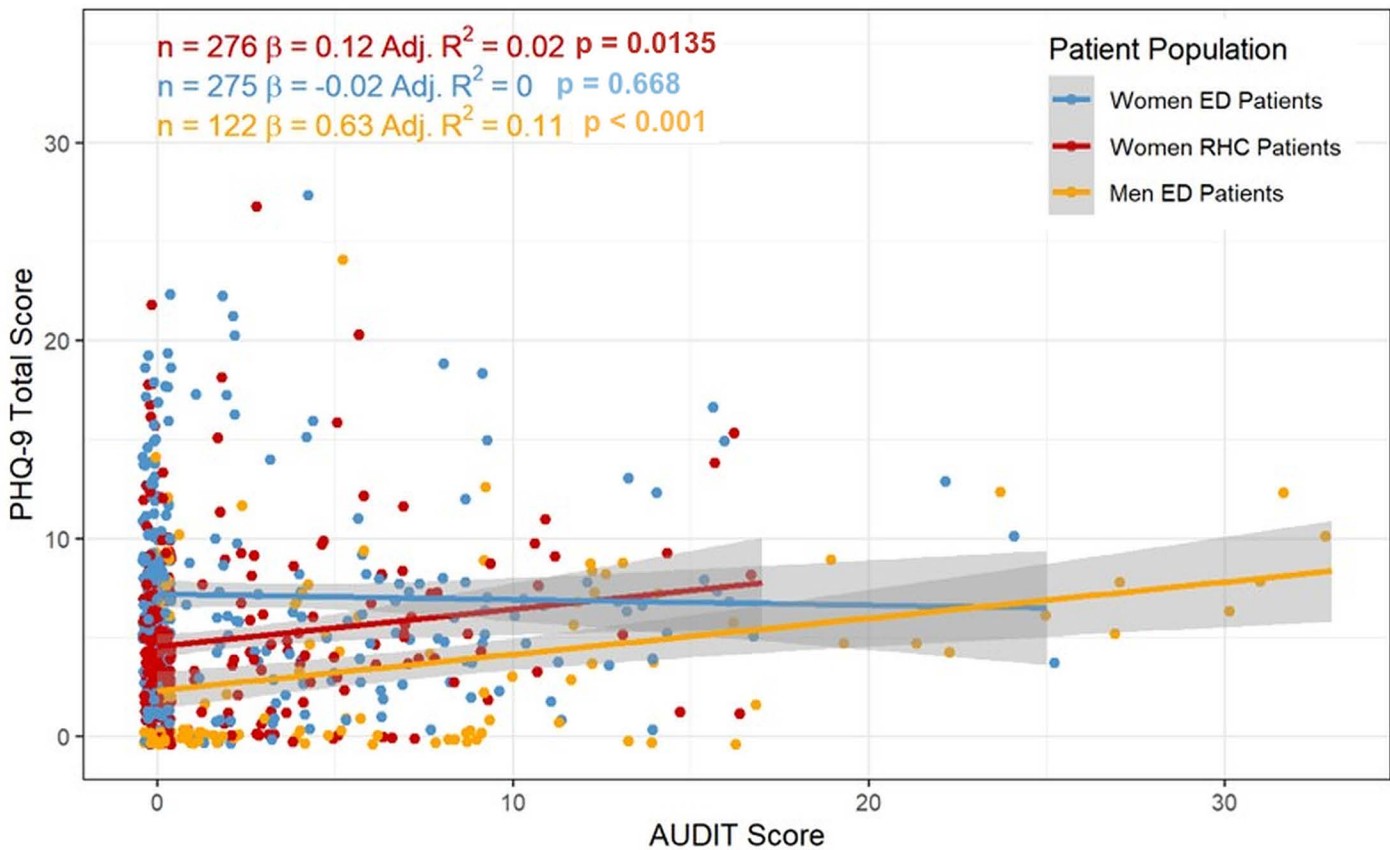

**Fig 1. Linear Regression between PHQ-9 and AUDIT Scores across Patient Populations.**

**Table 4. Qualitative Themes and Sub-Themes.**

| Themes | Sub-Themes |
| --- | --- |
| Relationship between alcohol use and mental well-being | Alcohol use and financial stress |
| | Alcohol use as a coping mechanism for stress |
| Gender imbalances in alcohol use's effect on mental well-being | Lack of healthy outlets for stress among men |
| | Women's social networks as a protective factor |

relative lack of support for men impacted their alcohol consumption behaviors and negatively influenced their overall mental health.

**Relationship between alcohol use and mental well-being.** Alcohol use as a contributor to financial stress: Respondents reported that alcohol consumption exacerbated life stressors, mentioning an increase in monetary stress among both genders, but especially in men. Exemplifying this, one respondent highlighted the financial stress created in his own life by his overconsumption of alcohol and how it affected his mental state and his family's financial opportunities.

*"I was drinking too much. We used to start drinking from 5pm in the evening until 1am at night! And you spend more than 200,000-300,000 shillings per night, and the wife does not know! So, we used to drink strong spirits like Konyagi, 3-4 big bottles per night. Thus, that time of my life was so boring and miserable. That habit put me under stress and depression for about a year. My life got back to zero. I couldn't afford even paying school fees for my kids."* (IDI #12, PHQ-9 = 7)

From this time spent drinking, he concluded that "*alcohol never reduced stress.*" Rather, because of his drinking, his "*stress was even worse*". Another respondent described how men "*spend all the money*" on consuming alcohol "*and come back home with nothing*" (IDI #11, PHQ-9 = 1), causing financial distress. A female participant shared:

*"There was a day I was stressed and I decided to go out for drinking and I ended up spending too much money, more than what I planned to spend. The next morning, I regretted everything a lot and I blamed alcohol."* (IDI #15, PHQ-9 = 15)

Not only did alcohol consumption lead to financial stress, but reciprocally, interviewees alluded to financial stress driving increased alcohol consumption. The anxiety that arose when men were unable to fulfill their perceived role as financial providers drove men to drink to lower their distress.

*"You may find a man working but not gaining enough to provide well for himself and his family, so he becomes so stressed as he is not respected as a provider. As a result, he falls into alcohol drinking in order to get rid of his problems."* (IDI #19, PHQ-9 = 5)

Alcohol use as a coping mechanism for stress: The cyclical pattern between alcohol consumption and financial stressors was mimicked throughout other interviews and illuminated the use of alcohol as a coping mechanism for stress. Participants widely suggested that alcohol is used to mitigate their own general stress or "*life hardship*" (IDI #17, PHQ-9 = 0) within their community when proper treatment was unavailable.

*"Many people who drink alcohol excessively use alcohol as a coping mechanism to handle or reduce stress and life difficulties such as losing a loved one or family problems."* (IDI #15, PHQ-9 = 13)

Another participant provided a personal experience of harmful alcohol use to cope with loss within their own family:

"*There was a time when my mom started unhealthy drinking after my dad passed away. She started that as a coping mechanism of losing my dad. It took her six months of unhealthy drinking. She was complaining about missing him so much and she was drinking a lot to the point where we had to carry her.*" (IDI #15, PHQ-9 = 15)

However, reliance on alcohol to reduce stress only instigated a repetitive chain of deleterious behaviors. Utilizing alcohol to relieve stress was described by one participant as "*pouring water in a jar that is leaking*" (IDI #17, PHQ-9 = 0). In other words, alcohol provided short-term relief but lacked any long-term benefit to psychological well-being or sustainable solutions to mental distress.

The negative impacts that resulted from the use of alcohol as a coping mechanism were explored by another participant, who shared:

"*When people face troubles they drink too much… [They] mistakenly use alcohol as a way to solve their problems. After becoming sober again they realize that the problems are still there. They get even more depressed because they have lost both time and resources on a wrong thing and find themselves drinking even more than before. The cycle goes on and on.*" (IDI #19, PHQ-9 = 5)

Overall, most participants had well-defined ideas regarding alcohol use as a coping mechanism for stress, life challenges, and depression, as well as the negative, cyclic impacts resulting from excessive use within their community.

**Gender imbalances in the effect of alcohol use on mental well-being.** The second central theme identified in IDIs was the gendered differences in how alcohol use impacts mental well-being and vice versa, with almost all participants agreeing that the effects were more severe among men.

"*Most men are affected more than women, because in my community men have a lot of responsibilities as compared to women. So, by the time he can't meet his goals, he ends up being frustrated and resorts to abusing alcohol and depression occurs.*" (IDI #17, PHQ-9 = 0)

While not necessarily normative, the lack of healthy outlets for men, strong social networks for women, and the unhealthy expectations of traditional male and female roles were all factors found to influence the relationship between alcohol consumption and mental well-being that disproportionately burdened men. Thus, highlighting the gender inequity in Tanzanian culture that is influencing alcohol consumption and mental well-being.

Lack of healthy outlets for men encourages alcohol intake: Throughout the interviews, many respondents highlighted a lack of healthy emotional outlets for men to deal with stressors or discuss their challenges – as said by one, "*a man never cries*" (IDI #16, PHQ-9 = 4), meaning that it was looked down upon socially for a man to express his emotions. This drove men to cope with stressful life situations and depression through alcohol:

"*Men only show a happy face even when they are stressed, because it is not easy for a man to talk about his problems. Many men drink alcohol to relieve their pain. I think a lot of men feel that showing their feelings of pain is a weakness, and that he will look like a weak man.*" (IDI #15, PHQ-9 = 13)

*"This problem affects more men than women in my society…They are always taught to keep the problems to themselves as a sign of strength or manhood. That's why, when problems overwhelm them, they get into depression. They can't tell anyone so they get into drinks and drink alone."* (IDI #16, PHQ-9 = 4)

The phrase "a man never cries" and the social significance of being labeled a weak man, according to another respondent, created more mental distress and depression among men compared to women. This burden was noted to be lesser on women because they were socially allowed to express feelings of weakness and hardship. One participant labeled these cultural views of masculinity as "toxic", and called for broader societal changes, such as creating new norms for men to be more open and honest about their challenges.

*"You know, naturally men are meant to be strong. Sometimes men experience problems, but you will never see them sharing to a fellow man, you know why? That's a sign of weakness. And this is the toxic feelings that we as men create every day, contrary to women who like to share anything they are going through. Through sharing you are releasing all the pains and sorrow and bad feelings… so the heart feels fresh and strong! I think this is the main reason why men get depressed compared to women."* (IDI #11, PHQ-9 = 1)

It was not a lack of desire, but rather a sociocultural norm that meant men were unable to share emotionally with other men. One respondent expressed his desire to talk about personal troubles, but found he is only able to do so when he and his friends are intoxicated and there is little chance of the event being remembered:

*"When I am stressed and I go out with my friends and we are drunk, I will open up all my problems because I am sure the next day they will not remember what I shared with them. I see it's a good way to let everything out as I am sure the next day everything will be forgotten."* (IDI #15, PHQ-9 = 15)

The desire to share his hardships only when his social network will forget further illuminates the overall lack of socially acceptable healthy outlets for men to discuss their mental well-being.

Women's social networks: In stark contrast to men's inability to share with other men, robust social networks among many women in Moshi were seen as a protective factor in combating stress – *"I think men are more stressed and they hide their stress on alcohol use. Women, when they are stressed, talk a lot and that helps them to reduce stress"* (IDI #15, PHQ-9 = 15). It was seen as easy and commonplace for women to talk with one another, and alcohol was not needed to facilitate these kinds of conversations – *"A woman may drink one beer and start to open up all her problems, while it takes a man to drink even ten beers and he still may not open up about his problems"* (IDI #15, PHQ-9 = 15). These social networks among women were so strong that women who were strangers still came to each other for help.

*"When women have a problem, they talk about it. That helps to take their pain away. A woman can meet anyone, even if she doesn't know her, and talk to her to relieve her heartache. This helps many women to cope with stress."* (IDI #9, PHQ-9 = 13)

In summary, participants proposed that depression and harmful alcohol consumption are strongly influenced by cultural norms. One primary effect of local norms is reduced autonomy and financial disenfranchisement among women, which in turn influence their rates of

depression and alcohol use. Another primary effect is the increased pressure placed on men to be the financial caretakers for their families. Social norms mediate that in the face of this stress, men are expected to "'*keep problems to themselves as a sign of strength*" and "*find solutions all by yourself,*" while women in the community are encouraged to talk to other women and find it easy to find "*a listening ear*" (IDI #16, PHQ-9 = 4). In effect, women are shown to employ healthier coping strategies while men utilize unhealthy coping strategies in the form of alcohol use.

## Discussion

This study, which is among the first of its kind in Northern Tanzania, investigated the intersection between gender, alcohol use, and depression among ED and RHC patients at KCMC. Existing literature within Moshi has identified high levels of alcohol use, described depression among specific patient populations, and explored gender differences in alcohol consumption and alcohol-related consequences [26,27,32–36,38,56]. Our mixed-methods analysis builds upon this literature to help inform the design and implementation of more effective mental health-related treatment options in this region.

Key findings from this analysis were that ED women patients had the highest rates of MDD among the three populations studied, indicating that this group may be particularly in need of more concerted clinical mental health care efforts. Importantly, though, an increasing severity of depressive symptoms in women did not correlate with increased alcohol use. This observation contrasts with ED men, namely, that there was a positive correlation between AUD and depressive symptoms. This relationship is supported by our qualitative data, which demonstrate that men in Moshi lacked social outlets for dealing with stress, making them more likely to turn to alcohol as a coping mechanism. It appears that social norms disenfranchise women by limiting their autonomy and financial opportunities, which may contribute to higher rates of depression among women. Men, by contrast, face cultural norms of being the main financial providers, which contributes to increased stress. However, in the face of these differing stressors, women employ healthier social coping strategies, while men are more likely to remain silent and turn to alcohol. These findings illustrate the need to take both a cultural and gender lens to mental health treatment in Moshi, utilizing social networks and differentiating interventions by gender to optimize treatment and health outcomes. Interventions might also examine whether social norms and gendered coping strategies reflect underlying values and goals or should be reconsidered to better align with these values. For example, previous work based in Kenya has shown the downstream benefits for family and community members when interventions that address both alcohol use and depression are implemented [57–59].

Our results show that ED women have the highest rates of MDD among all populations, highlighting the need for mental health interventions to better support this high-risk group. This reflects global differences in the prevalence of depression among men and women, with depression being 50% more common in women compared to men [60]. Paralleling our estimates, in a systematic review of 134 studies, Corcoran et al. found the prevalence of depression among women in LMICs was almost twice as high among women living in higher-income countries [61]. Within our qualitative data, many interviewees described depressive symptoms without ever stating or naming depression specifically. The hesitancy to label themselves or others as depressed could highlight the mental health stigma that is present in Tanzania and globally [62,63].

Although our results demonstrate higher rates of depression among women, depression was not associated with increased alcohol use among women. Interestingly, this was not true for male participants where depressive symptoms were positively and linearly associated

with alcohol consumption. This dichotomy suggests that men's and women's approaches to alcohol use and stress relief differ significantly. Stress has been widely associated with alcohol use, with previous research conducted by our team identifying stress as a major motivator for unhealthy drinking habits in Moshi [32]. A major underlying factor in gender differences between stress and alcohol use is the influence of social networks. Our qualitative findings suggest a higher prevalence of strong social support networks for women, which may be associated with low rates of alcohol use and harmful drinking behaviors. This link is also seen in global data, where social support networks have been seen to offer a protective effect on depression risk [64].

As was noted within our IDIs, the relative absence of emotional support for men may lead them to use alcohol as a coping mechanism for difficult events and thus calls for an increase in healthy emotional outlets for men. While social support networks may be helpful for women experiencing emotional distress, it is still important to acknowledge the larger clinical burden of depression experienced by women in future mental health interventions within this setting. Particularly, previous research has examined the disproportionate stigma experienced by women who consume alcohol in Tanzania [65]. A study recently conducted in South Africa suggests that increasing social support may not be enough to prevent depression among women who are experiencing high rates of stigma [66]; therefore, it may be necessary to incorporate additional targeting counseling and advocacy focused on addressing harmful cultural norms and gender inequities that contribute to the greater risk of depression among Tanzanian women. Future mental health-related interventions in this region must be cognizant of cultural norms and resulting behavioral differences underlying mental health disorders to more efficiently and effectively distribute available resources.

The findings discussed thus far highlight the need for gender and sociocultural factors to be more closely considered in mental health-related programs and, more specifically, for there to be greater differentiation in interventions intended for men versus women. For women, the high prevalence of MDD identified in this paper, especially among those receiving care at the ED, points to a need for more robust mental health-related screening, referral, and care services for female patients. Previous work by our team has highlighted the relevance of expanding these services within KCMC's ED [67]. In the context of this paper, it is essential to note the potential benefit this could have among women given their significant mental health burden.

In contrast, for men, the linear relationship between alcohol use and depressive symptoms in this analysis indicates that local alcohol-related interventions may benefit from teaching and providing healthy, values-based coping mechanisms when faced with life stressors. Mental health-related treatments that are differentiated by gender have been shown to have greater effectiveness [68–70] and individualized treatments for men may have wider cultural acceptance.

Previous research and clinical trials have also shown the value of incorporating a multi-level approach to mental health and substance use treatment, addressing individual-, family-, and community-level interventions. Mathias et al. found that incorporating multi-pronged elements to care delivery, including increased access to services, greater education around mental health topics, and robust social support networks, was effective in treating patients with severe mental health problems in India [71]. In Tanzania specifically, multi-level approaches to mental health care are gaining support [72] and have been locally implemented in the treatment of other medical issues like HIV [73]. As such, we recommend that local health policy efforts incorporate these considerations in mental health delivery efforts to provide more effective care for the Moshi community.

## Strengths and limitations

Though this paper offers novel insights into the mental health landscape of Moshi, the data presented here must be considered along with some broader limitations. In terms of qualitative data, specific interview questions on depression and alcohol use were only added midway through data collection, once the research team recognized the importance of this topic in preliminary analyses. While added later on, saturation on the topic of depression and alcohol use was still reached and was done so using robust qualitative data collection procedures, including gender matching of interviewers and interviewees, rigorous qualitative training among the research team, and ensured privacy and confidentiality. Therefore, we hold that the IDI data presented here is representative and accurate for the study population.

In addition, some key drivers of MDD such as past trauma, medical conditions, and genetic factors were not measured in the quantitative survey. Information like this could help identify factors within the broader population that should be addressed in community outreach programs. Therefore, we encourage future research to explore the reasons underlying depressive symptoms in this region, especially in the ED women population that was identified as high-risk. Even in light of these limitations, this study adds meaningfully to the existing literature, standing as the first study to specifically explore the intersection of alcohol use, depression, and gender within Moshi and can serve as a foundation for future related research.

## Conclusion

Women in Tanzania are more likely to experience major depressive disorder. Depression was not associated with higher alcohol use among women but was significantly associated with increased alcohol use among men. Women experiencing depression were more likely to rely on healthy coping through emotional support networks, whereas these networks were weaker for men, which may contribute to men using unhealthy strategies such as drinking alcohol when experiencing emotional distress. Our findings highlight the need for future mental health-related interventions in this region to address harmful gender norms, to take gender into greater account in the design of interventions, and to include multiple aspects of care delivery designed to address individual, family, and community aspects of overall patient well-being.

## Supporting information

**S1 Checklist. STROBE Checklist.**
(DOCX)

**S1 Questionnaire. PLOS Inclusivity in Global Research Questionnaire.**
(DOCX)

## Author contributions

**Conceptualization:** Alena Pauley, Judith Boshe, Blandina T. Mmbaga, Catherine A. Staton.

**Data curation:** Alena Pauley, Yvonne Sawe, Mariana Mikindo, Joseph Kilasara.

**Formal analysis:** Judith Boshe, Blandina T. Mmbaga, Catherine A. Staton.

**Funding acquisition:** Alena Pauley, Catherine A. Staton.

**Investigation:** Judith Boshe, Blandina T. Mmbaga, Catherine A. Staton.

**Methodology:** Alena Pauley, Kirstin West, Catherine A. Staton.

**Supervision:** Sharla Rent, Blandina T. Mmbaga, Catherine A. Staton.

**Writing – original draft:** Alena Pauley, Mia Buono, Madeline Metcalf, Kirstin West, William Nkenguye.

**Writing – review & editing:** Alena Pauley, Sharla Rent, Brandon A. Knettel, Catherine A. Staton.

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
