## [Decision Letter · Decision Letter 0]

20 Feb 2024

PGPH-D-23-02200

“A Man Never Cries”: A Mixed-Methods Analysis of Gender Differences in Depression and Alcohol Use in Moshi, Tanzania

Dear Dr. Staton,

Thank you for submitting your manuscript to PLOS Global Public Health. After careful consideration, we feel that it has merit but does not fully meet PLOS Global Public Health’s publication criteria as it currently stands. Therefore, we invite you to submit a revised version of the manuscript that addresses the points raised during the review process.

We look forward to receiving your revised manuscript.

Kind regards,

Ikenna D Ebuenyi, MBBS,PhD

Academic Editor

Journal Requirements:

Additional Editor Comments (if provided):

Thank you for submitting your manuscript for review. Although the reviewers acknowledge the relevance of the manuscript, they have identified some flaws in the methodology, results description and presentation. The reviewers have provided useful recommendations that could improve the manuscript. We invite you to carefully consider and address the reviewers’ comments and recommendations and submit a revised manuscript.

Reviewers' comments:

Reviewer's Responses to Questions

**Comments to the Author**

1. Does this manuscript meet PLOS Global Public Health’s publication criteria ? Is the manuscript technically sound, and do the data support the conclusions? The manuscript must describe methodologically and ethically rigorous research with conclusions that are appropriately drawn based on the data presented.

Reviewer #1: Yes

Reviewer #2: Yes

2. Has the statistical analysis been performed appropriately and rigorously?

Reviewer #1: No

Reviewer #2: Yes

3. Have the authors made all data underlying the findings in their manuscript fully available (please refer to the Data Availability Statement at the start of the manuscript PDF file)?

Reviewer #1: No

Reviewer #2: Yes

4. Is the manuscript presented in an intelligible fashion and written in standard English?

Reviewer #1: Yes

Reviewer #2: Yes

5. Review Comments to the Author

Reviewer #1: PGPH-D-23-02200

Follow a reporting guideline such as STROBE so that no comments are missing in reporting.

Please report the reliability and validity of the tool that the author used.

The questionnaire development and validation are required to be reported.

Stressing PHQ-9 and AUDIT throughout the manuscript creates ambiguity, so it is recommended to use depression.

The statistical model requires an explanation of why it was used.

All the assumptions must be met before fitting the model. And they must be reported.

If separate models such as linear regression and logistic regression were used, their assumptions would not be the same. The RR as an estimate needs to be checked, as the estimate from the binary regression is the odds ratio.

All the models' fitness should be reported. The model's fitness can be plotted. The plots are different for each model. Please report the plots.

Please address the comments and reports as suggested.

Reviewer #2: A Man Never Cries”: A Mixed-Methods Analysis of Gender Differences in Depression and Alcohol Use in Moshi, Tanzania

Line Present Comment

75 No key word To add key word

136 Method To be summarized in proper way

300 Relation ship To use correlation

301 Linear relation Is it positive relation or negative relation

310 Linear relation figure Not available

469 Our Better to avoid use our, we

489 Our results Not use our

578 Reference What is style

581 Reference What is style

587 Reference What is style

6. PLOS authors have the option to publish the peer review history of their article (what does this mean? ). If published, this will include your full peer review and any attached files.

**Do you want your identity to be public for this peer review?** For information about this choice, including consent withdrawal, please see our Privacy Policy .

Reviewer #1: No

Reviewer #2: No

---

## [Decision Letter · Decision Letter 1]

25 Sep 2024

PGPH-D-23-02200R1

“A Man Never Cries”: A Mixed-Methods Analysis of Gender Differences in Depression and Alcohol Use in Moshi, Tanzania

Dear Dr. Staton,

Thank you for submitting your manuscript to PLOS Global Public Health. After careful consideration, we feel that it has merit but does not fully meet PLOS Global Public Health’s publication criteria as it currently stands. Therefore, we invite you to submit a revised version of the manuscript that addresses the points raised during the review process.

The revised manuscript has been assessed and reviewers have pointed out some further necessary revisions, please review their comments below and make all of the necessary revisions to address these concerns. Please specifically include the sample size calculation and inclusion criteria in the methodology.  

We look forward to receiving your revised manuscript.

Kind regards,

Emma Campbell, Ph.D

Staff Editor

Journal Requirements:

Reviewers' comments:

Reviewer's Responses to Questions

**Comments to the Author**

1. If the authors have adequately addressed your comments raised in a previous round of review and you feel that this manuscript is now acceptable for publication, you may indicate that here to bypass the “Comments to the Author” section, enter your conflict of interest statement in the “Confidential to Editor” section, and submit your "Accept" recommendation.

Reviewer #2: All comments have been addressed

Reviewer #3: (No Response)

2. Does this manuscript meet PLOS Global Public Health’s publication criteria ? Is the manuscript technically sound, and do the data support the conclusions? The manuscript must describe methodologically and ethically rigorous research with conclusions that are appropriately drawn based on the data presented.

Reviewer #2: (No Response)

Reviewer #3: Partly

3. Has the statistical analysis been performed appropriately and rigorously?

Reviewer #2: Yes

Reviewer #3: No

4. Have the authors made all data underlying the findings in their manuscript fully available (please refer to the Data Availability Statement at the start of the manuscript PDF file)?

Reviewer #2: Yes

Reviewer #3: Yes

5. Is the manuscript presented in an intelligible fashion and written in standard English?

Reviewer #2: Yes

Reviewer #3: Yes

6. Review Comments to the Author

Reviewer #2: thanks for your efforts

Reviewer #3: Line 140: The type of study design for the survey is to be stated. For cross-sectional study design, RR is not suitable to be used unless if the study outcome is rare (OR can be interpreted similarly to RR). In this case unlikely.

Line 151/253: What if the sample size of 20 is not sufficient for research study to reach saturation?

Line 153: Inclusion criteria is to be stated.

Line 206: Analysis is to be replaced with Statistical Analyses.

Line 213: Studying the relationship between AUDIT and PHP-9 alone is not sufficient. The outcome can be contributed by other/confounding factors.

Line 215: Can the relationship be the other way round e.g. AUDIT as dependent and PHQ-9 as predictor or a bilateral relationship?

Line 230: What about other variables like age, employment, marital status?

If the variables in the unadjusted analyses were found to be significant, adjusted analysis could be done.

Line 231: The version of the software is to be mentioned.

Line 232: Which variable that was adjusted using Mantel-Haenszel is to be mentioned.

Line 239: The acceptance significance level of p value, one or two-tailed test p value is to be stated.

For qualitative method, please state if all the interviews were conducted face to face. Were the interviews recorded?

Line 292-293: The reference number is to be provided.

Table 1, 2, 3: The statistical tests employed here is to be denoted in table footnote. If there are additional statistical tests used for these tables, it is to be mentioned in the statistical analysis section (Line 206). For Mantel-Haenszel test, indicate which variable was adjusted.

For the results section, the decimal point for percentage figures in the text is to be at least 1 decimal point and consistent.

P Value to be replaced with p= or p<

Line 325-329: The word correlation refers to r (correlation coefficient). R² (coefficient of determination) does not directly represent the correlation but rather how much of the variance in PHQ-9 scores is explained by AUDIT scores. If correlation is reported, r instead of R² is to be used.

For Table 3: the frequency for predictors to be presented before deriving the RR.

Missing data information is to be provided/stated if any.

For the qualitative analysis, the study ID of the participants could be used to indicate their responses.

Revisit the STROBE checklist and ensure all the information stated there are highlighted in the manuscript.

7. PLOS authors have the option to publish the peer review history of their article (what does this mean? ). If published, this will include your full peer review and any attached files.

**Do you want your identity to be public for this peer review?** For information about this choice, including consent withdrawal, please see our Privacy Policy .

Reviewer #2: **Yes: ** Safa A faraj MD

Reviewer #3: No

---

## [Decision Letter · Decision Letter 2]

3 Jan 2025

“A Man Never Cries”: A Mixed-Methods Analysis of Gender Differences in Depression and Alcohol Use in Moshi, Tanzania

PGPH-D-23-02200R2

Dear Dr. Staton,

We are pleased to inform you that your manuscript '“A Man Never Cries”: A Mixed-Methods Analysis of Gender Differences in Depression and Alcohol Use in Moshi, Tanzania' has been provisionally accepted for publication in PLOS Global Public Health.

The reviewer had one remaining comment; I believe this can be addressed during your final formatting checks.

Best regards,

Julia Robinson

Executive Editor

Reviewer Comments (if any, and for reference):

Reviewer's Responses to Questions

**Comments to the Author**

1. If the authors have adequately addressed your comments raised in a previous round of review and you feel that this manuscript is now acceptable for publication, you may indicate that here to bypass the “Comments to the Author” section, enter your conflict of interest statement in the “Confidential to Editor” section, and submit your "Accept" recommendation.

Reviewer #3: All comments have been addressed

2. Does this manuscript meet PLOS Global Public Health’s publication criteria ? Is the manuscript technically sound, and do the data support the conclusions? The manuscript must describe methodologically and ethically rigorous research with conclusions that are appropriately drawn based on the data presented.

Reviewer #3: Yes

3. Has the statistical analysis been performed appropriately and rigorously?

Reviewer #3: No

4. Have the authors made all data underlying the findings in their manuscript fully available (please refer to the Data Availability Statement at the start of the manuscript PDF file)?

Reviewer #3: Yes

5. Is the manuscript presented in an intelligible fashion and written in standard English?

Reviewer #3: Yes

6. Review Comments to the Author

Reviewer #3: For figure 1, symbol = is missing from the figure i.e. p 0.0135, p 0.668

Line 355-356: p=0.668 p=0.014

7. PLOS authors have the option to publish the peer review history of their article (what does this mean? ). If published, this will include your full peer review and any attached files.

**Do you want your identity to be public for this peer review?** For information about this choice, including consent withdrawal, please see our Privacy Policy .

Reviewer #3: No
